# Why Protect Decapod Crustaceans Used as Models in Biomedical Research and in Ecotoxicology? Ethical and Legislative Considerations

**DOI:** 10.3390/ani11010073

**Published:** 2021-01-03

**Authors:** Annamaria Passantino, Robert William Elwood, Paolo Coluccio

**Affiliations:** 1Department of Veterinary Sciences, University of Messina-Polo Universitario Annunziata, 98168 Messina, Italy; 2School of Biological Sciences, Queen’s University, Belfast BT9 5DL, Northern Ireland, UK; R.Elwood@qub.ac.uk; 3Department of Neurosciences, Psychology, Drug Research and Child Health (NEUROFARBA), University of Florence, Viale Pieraccini 6, 50139 Firenze, Italy; paolo.coluccio@unifi.it

**Keywords:** decapoda, model organism, ecotoxicology, 3Rs, animal welfare legislation

## Abstract

**Simple Summary:**

Current European legislation that protects animals used for scientific purposes excludes decapod crustaceans (for example, lobster, crab and crayfish) on the grounds that they are non-sentient and, therefore, incapable of suffering. However, recent work suggests that this view requires substantial revision. Our current understanding of the nervous systems and behavior of decapods suggests an urgent need to amend and update all relevant legislation. This paper examines recent experiments that suggest sentience and how that work has changed current opinion. It reflects on the use of decapods as models in biomedical research and in ecotoxicology, and it recommends that these animals should be included in the European protection legislation.

**Abstract:**

Decapod crustaceans are widely used as experimental models, due to their biology, their sensitivity to pollutants and/or their convenience of collection and use. Decapods have been viewed as being non-sentient, and are not covered by current legislation from the European Parliament. However, recent studies suggest it is likely that they experience pain and may have the capacity to suffer. Accordingly, there is ethical concern regarding their continued use in research in the absence of protective measures. We argue that their welfare should be taken into account and included in ethical review processes that include the assessment of welfare and the minimization or alleviation of potential pain. We review the current use of these animals in research and the recent experiments that suggest sentience in this group. We also review recent changes in the views of scientists, veterinary scientists and animal charity groups, and their conclusion that these animals are likely to be sentient, and that changes in legislation are needed to protect them. A precautionary approach should be adopted to safeguard these animals from possible pain and suffering. Finally, we recommend that decapods be included in the European legislation concerning the welfare of animals used in experimentation.

## 1. Introduction: Decapod Crustacean as Models

Vertebrate and invertebrate models are indispensable tools for studying and understanding biological and ecological processes. Commonly used examples are the invertebrates *Caenorhabditis elegans* and *Drosophila melanogaster*, and the vertebrates *Mus musculus*, *Rattus norvegicus, Danio rerio*, *Xenopus laevis* and *Gallus gallus domesticus* [1,2,3]. However, there is current pressure to cut the use of vertebrates, and to use invertebrates instead, to partly satisfy one criterion of the “3Rs” principles (Replacement) [4] in animal welfare [5,6,7]. This requires an evaluation of sentience so that an animal with “higher” sentience is replaced by one that is “lower” (often unprotected by legislation, i.e., embryos of vertebrates before of the last third of their normal development or invertebrates except live cephalopods). To this end, the EU Directive 63/2010 [8] states that animals “*with the lowest capacity to experience pain, suffering, distress or lasting harm”*, should be used.

Invertebrates vary enormously in neural and behavioral complexity [9,10]. Some have diffusely organized nervous systems with relatively small numbers of neurons (such as nematodes), and for these there is little concern for their welfare. The taxa that give rise to most welfare concerns are the arthropods (e.g., crustaceans, insects) and some molluscs (e.g., cephalopods). These invertebrates have large, complex nervous systems containing a very high number of neurons, and varied behavior, which may be modified by early social conditions [11]. The cephalopods receive protection [8], but this is not extended to other invertebrate groups.

Recently, decapod crustaceans have become important models for biochemical, physiological, and ecological research [12,13,14,15,16], due to their biological characteristics and/or their convenience of collection/use. These characteristics include advanced circulatory hormones, immune systems, ease of culture, suitable size, individual traits, tolerance to handling, high fertility, relatively short generation time, and adaptability to a wide spectrum of environmental and nutritional conditions [17,18,19]. The crayfish, *Procambarus clarkii,* is used in physiological and immunological studies and as a biomarker in contaminated ecosystems [20,21,22,23,24,25,26,27,28]. Another crayfish used for a variety of studies (microscopic anatomy, physiology, development, behavior, toxicology) is *Procambarus virginalis* [29,30,31,32,33,34,35,36,37,38,39,40,41,42,43], which is a vigorous, widely distributed eurytopic species [44]. It is parthenogenetic and produces numerous genetically identical offspring that can be used in many areas of research [44]. The freshwater prawn, *Macrobrachium* spp., and some crabs, e.g., *Chasmagnathus* spp. and *Portunus* spp., are commonly used in toxicologic studies [45,46]. Further, there is a vast range of species-specific studies on all aspects of biology that are performed on other species of decapods.

There are no minimal animal care requirements for decapods in the European legislation [47]. They have long been regarded as not being sentient and unable to experience pain or suffering because they were thought to respond to noxious stimuli purely by nociceptive reflex [48,49]. Nevertheless, in 2005 the Animal Health and Animal Welfare Panel of the European Food Safety Authority [50] recommend their inclusion in the legislation that provides protection for animals. However, after consideration, the decapods were excluded from the subsequent 2010 EU Directive due to the refutation of the conclusiveness of the studies cited by the panel. Here we examine that position in the light of recent studies that examine if decapods might show responses to noxious stimuli that are consistent with the idea that they experience pain. We then consider if they should now be included in welfare guidelines and legislation, as applied to vertebrates and cephalopods in experimentation.

## 2. Pain, Stress and Potential Suffering in Decapod Crustaceans

Pain is a complex phenomenon involving sensory, cognitive and affective components. When the pain is not mitigated, it can produce distress, suffering and harmful effects to physical health, and thus is a welfare issue [51]. The International Association for the Study of Pain (IASP) has recently defined pain as “*An aversive sensory and emotional experience typically caused by, or resembling that caused by, actual or potential tissue injury”*, [52]. This concentrates on the internal emotional state, which is central to our concern for welfare, but that internal state cannot be accessed by researchers. By contrast, Elwood [53] defined pain as “*a non-reflexive response to a noxious, potentially tissue-damaging stimulus that alters future behaviour*”. The focus here is on aspects that can be measured within appropriate experimental studies. Non-reflexive responses are important because, for many years, the idea of pain in decapods was dismissed given that it was thought that all responses to noxious stimuli were reflexive, and there was no need to invoke ideas of pain. The rationale is that virtually all animals have neurons (nociceptors) that respond to tissue damage, mechanical stimuli, and noxious chemicals. In many cases these allow for a withdrawal response via a reflex loop, and without signals going to the centralized neuronal masses or brain. Thus, there is no feeling or noxious experience. However, if responses are shown to go beyond reflexes it opens the possibility that they are mediated by pain [54]. Further, we consider the alteration of future behavior because pain must serve a function by enhancing recovery from damage and reducing future damage [10]. These and other aspects have been the focus of a series of experiments in crustaceans that were published after the 2005 EU recommendations for decapod protection.

Information about animal pain might be gained from studies using the following criteria:(i)*Protective motor reactions*. Vertebrates often show limping, rubbing, prolonged licking, and these are thought to indicate pain [55]. Similar responses have been noted in various decapod species. For example, when a single antenna of the glass prawn, *Palaemon elegans,* was brushed with 10% sodium hydroxide or 10% acetic acid, they showed prolonged rubbing of that specific antenna against the tank wall, and also vigorous grooming of the same antenna by repeatedly pulling it through the pincers on the front legs [56]. This shows an ability to localize the noxious stimulus on its body and is not a generalized reflex to stimulation. Further, shore crabs, *Carcinus maenas*, rubbed their mouth parts with their claws if brushed with acetic acid. Additionally, if one eye was brushed with acetic acid that eye was held down in the eye-socket for longer than if brushed with water, and that response was specific to the eye with acid [57]. Dyuzen et al. [58] injected formalin into an appendage of the crab, *Hemigrapsus sanguineus,* which then shook and rubbed the specific appendage, and reduced its use. Hermit crabs, *Pagurus bernhardus*, that had received electric shocks on the abdomen and had abandoned their shell showed grooming of the abdomen [59], and brown crabs that had one claw removed by twisting it off (as in fishery practice) picked at the wound and held their remaining claw over the wound when confronted by an intact crab [60]. Thus, decapods attend to the area of the body that received the noxious stimulation in a similar manner to that shown by mammals;(ii)*Trade-offs between avoidance responses and other motivational requirements*. Nociceptive reflexes should be the same irrespective of other motivational requirements [17,61]. If those responses vary according to other requirements, it would demonstrate central decision-making rather than reflex. That is, it would demonstrate a trade-off between avoidance of the noxious stimulus and another motivational priority [62]. Such motivational trade-offs were observed by Elwood and Apple [62] in hermit crabs (*Pagurus bernardus*) that were more likely to abandon a poor-quality shell after electric shock than one of high quality. They also emerged from low-quality shells at a lower intensity shock than did those in high quality shells [59]. Similarly, hermit crabs were less likely to leave their shells after electric shock if the odor of a predator was present in the surrounding water [63]. These studies demonstrate that getting out of a shell when shocked on the abdomen is not a reflex because the avoidance of the electric shock is traded-off against maintaining a high-quality shell or the avoidance of predators;(iii)*Long-term motivational changes.* If the animal shows long-lasting changes in behavior, then that change cannot be described as a reflex. This was noted when hermit crabs were shocked within their shells. Although some evacuated from the shell, many did not, and these were subsequently offered a new shell. These shocked crabs were more likely than non-shocked crabs to approach the new shell and move into it [62,64]. They did so quickly and with minimum investigation of the new shell, indicating that the shock had induced a high motivation to swap shells [65,66] that lasted at least 24 h [64];(iv)*Paying a cost to avoid the noxious stimuli.* If an animal pays a cost to avoid a noxious stimulus it demonstrates that the stimulus is highly aversive and that the animal strongly prefers to avoid it [10]. This was seen in hermit crabs that got out of their shell when the abdomen was shocked. Some crabs then moved away from the shell and some attempted to climb the wall of the observation chamber [59]. Whilst many hermit crabs reentered the shell, often after a prolonged investigation within, some remained naked for the 15 min observation period. Because shells are essential for survival for these hermit crabs, this indicates a high cost is paid to avoid the shock. Similarly, shore crabs emerged from a dark shelter if shocked within and entered a brightly lit area that is normally strongly avoided [67]. Crabs on the shore typically avoid predators by remaining hidden in dark crevices, so emerging indicates the aversive nature of the electric shock and the high costs paid to avoid it;(v)*Avoidance learning*. A key function of pain is that it should increase the salience of the noxious stimulus, thus making avoidance learning more likely. That is, the high motivation to avoid the pain should improve the learning and protect the animal from future damage. This was investigated by Magee and Elwood [67], by repeatedly placing shore crabs in the center of a brightly lit area that had two dark shelters. On the first trial all crabs quickly selected a shelter. Some crabs had been nominated to receive a shock in the first selected shelter, while the other crabs had been nominated to receive a shock only if they went to the alternative shelter on subsequent trials. On the second trial most crabs went to the shelter they had selected in the first trial and choice was not affected by prior shock. However, in the third trial significantly more shocked crabs switched their choice compared to those not shocked. That is, just two prior experiences overcame the apparent preference for one shelter and caused them to avoid that shelter. An earlier study by Kawai et al. [68] demonstrated that *P. clarkii* could associate the illumination of a light with a noxious electric shock given ten seconds later. Crayfish that faced the safe area learned to respond to the light by walking to the safe area and thus learned to avoid the shock. However, if it was facing away from the safe area the crayfish did not walk but responded to the shock by a tail-flick escape response that propelled it backwards and into the safe area. These animals did not learn to escape the shock by responding to the light; however, if they were subsequently turned around to face the safe area, they rapidly learned to walk when the light came on to avoid being shocked;(vi)*Anxiety.* An example of anxiety or increased wariness comes from work on crayfish [69]. Crayfish were tested in cross mazes in which two arms of the maze were brightly lit and the other two were dark. The crayfish used all parts of the maze but slightly more time was spent in the dark arms. However, some crayfish were exposed to repeated short-duration electric fields that induced escape responses, indicating that this treatment was noxious. These animals then spent far less time in the light arms of the maze than did those placed in the treatment area but without the shocks. That is, the normal preference for the dark was considerably enhanced and the shocked animals were described as showing ‘anxiety’. Animals exposed to shocks were then given the drug chlordiazepoxide, which is used to reduce anxiety in humans [70]. This seemed to reduce anxiety in the shocked crayfish because treated animals used the light arms as much as those that had not been shocked. Anxiety also makes amphipod crustaceans more cautious about potential danger, and thus improves survival in the presence of a predator [71];(vii)*Effects of analgesics and local anesthetics.* The rubbing and grooming by glass prawns of specific antennae brushed with noxious chemicals was markedly reduced if those antennae had been pretreated with a local anesthetic, such as benzocaine. That is, the benzocaine appeared to block the nociceptive input [72]. Further, lignocaine has been used to reduce aversive responses in the freshwater prawn, *Macrobrachium americanum*, to the aquaculture practice of eye-stalk ablation [72]. Thus, similar effects of local anesthetics are seen in decapods and vertebrates. Opioids reduce aversive responses to noxious stimuli in vertebrates such as fish [73], birds [74] and mammals [75]. Opioid peptides and receptors also occur in various invertebrates, but their analgesic effects are disputed [76,77,78]. Morphine reduced responses to electric shocks in crabs, *Chasmagnathus granulatus*, [79,80], and this was reversed by the opioid antagonist, naloxone. However, morphine also reduced the escape response to a moving shadow of the crab, *C. granulatus* [81], suggesting that the apparent analgesic effects of morphine were simply due to reduced responsiveness to all stimuli [81].

This possibility was tested by Barr and Elwood [78] using the shore crab *Carcinus maenas.* Crabs were either given morphine or water injections and placed in a light area that had a single dark shelter into which the crabs moved. Some crabs received an electric shock within the shelter, but others did not. Each crab was tested for twenty trials after a single injection, and whether they entered the shelters and the latency to enter were recorded. The rationale was that if morphine had an analgesic effect then more crabs should move into the shelter when paired with a shock compared to those without morphine. However, this was not found. Irrespective of shock or not, crabs given morphine showed low numbers of shelter entries during the first ten trials and appeared to be unresponsive and limp. They soon recovered, however, and in the second ten trials there was no difference in shelter entry between those given morphine and those given water injections. This supports the idea that the suggested analgesic effects noted in other studies were simply due to a general lack of response rather than analgesia [78].

(viii)*Physiological responses.* Pain activates various physiological responses, such as modifications in heart rate, respiration and/or hormonal levels [17,73,82,83], and these are generally regarded as stress responses. Stress is a biological response that an animal exhibits to cope with a threat to its homeostasis [84]. It occurs in vertebrates when environmental conditions are outside of their normal physiological range, or from aversive stimuli such as a those from predators. Initially, stress is adaptive (eustress) and enhances the ability to cope in the short term. However, if it persists the stress has negative impacts on important functions. Vertebrate stress can be assessed by measuring the hormonal and biochemical state [85]. Generally, it causes a cascade of hormonal changes that leads to the production of adrenal hormones (cortisol and corticosterone), which cause the conversion of glycogen to glucose for use in the flight or fight response. In decapods, the stress causes an increase in crustacean hyperglycemic hormone (CHH) and/or a release of biogenic amines such as epinephrine and serotonin [86,87]. These control a variety of physiological processes and function similarly to the corticosteroids in vertebrates, in that glycogen is converted to glucose. They also cause elevated lactate [88,89], which has been used as a proxy for measuring stress [90,91,92,93]. Removing one claw of edible crabs by twisting the claw had rapid physiological effects [94]. When compared to control animals there was a significant increase in lactate and glucose, and a marked shift in the glucose to glycogen ratio. This was not seen in crabs induced to autotomize [94], suggesting that the effects were predominantly due to the tissue damage caused by manual declawing. Similar physiological changes occur in response to a wide variety of conditions [94,95], but a key question is do they occur in response to stimuli that might cause pain?

Crayfish exposed to repeated electric shocks respond by repeated tail-flick escape responses and elevated dopamine and serotonin [69,70]. The latter has been shown to be involved in mediating the anxiety response noted above. Shocked crayfish had elevated serotonin (5HT) levels in the brain and non-shocked animals injected with 5HT showed similar levels of anxiety to those that were shocked. Crayfish, pre-treated with a 5HT agonist, did not show the anxiogenic effect of 5HT [70]. There were also close correlations between 5HT levels and behavioral indicators of anxiety. Serotonin also causes the release of the crustacean hyperglycemic hormone, which elevates hemolymph glucose concentrations and lactate [94,95,96].

However, it is important to consider a potential confounding variable when considering stress hormones in response to noxious stimulation. Often, noxious stimulation, such as electric shock, causes vigorous escape behavior, and this behavior, rather than the shock per se, might cause at least some physiological change [65]. For example, in the studies of Fossat et al. [69,70], crayfish were subject to electric charges repeated at 5 s intervals over a period of 30 min, which caused vigorous tail-flipping escape responses. When the power of the electric charges was reduced, and flipping was not observed, there was no physiological change. At higher shock intensities, the tail flipping decreased over time and this was suggested to be due to habituation [70]. It could, however, have declined due to exhaustion, induced by high lactate. One study specifically addressed this problem in shore crabs. These were shocked at 10 s intervals for 2 min, i.e., less frequent and for a much shorter time than for the crayfish [97]. Shore crabs do not show tail-flicking responses to shock but some show escape responses by attempting to climb the walls of the tank, and others showed threat responses. However, many shocked crabs did not show vigorous responses and simply walked. Control crabs were not shocked and a few of these remained motionless, whereas most walked in the test tank. The key comparison in this experiment involved those shocked crabs and those control animals that showed the same behavior, i.e., that walked. Lactate was significantly higher in shocked than non-shocked crabs, thus showing that the physiological stress response was caused by the noxious stimulus rather than the behavior that it elicited. The implication of this is that the shock was sufficiently aversive to induce a stress response in a similar way to that seen in vertebrates in pain.

We have focused here on key behavioral and physiological responses to noxious stimuli that are consistent with the idea of pain. Sneddon et al. [10] considered 17 specific criteria (Table 1) that, taken together, can provide evidence for animal pain. Note that our category of “anxiety” is included within category 13 from Sneddon et al. [10], i.e., “altered behavioral choices/preferences”. Table 1 shows that of the 17 criteria, only three are not fulfilled for decapods. However, that is not because the experiments failed to show self-administration of analgesia, or paying a cost to access analgesia, or relief learning; rather, no such experiments have been reported. Thus, decapods fulfill all 14 criteria for which they have been tested. We accept that it is not possible to prove beyond all doubt that any animal species has the capacity to experience pain. Nevertheless, we note that, considering recent experimentation, there is no compelling argument to dismiss the idea of pain in decapods. This is particularly because the prior argument that they only respond by nociceptive reflexes, and thus cannot feel pain, has been shown to be false. Thus, we must accept the possibility of pain and sentience in this taxonomic group.

## 3. Ethical Issues and Recent Concerns

The maxim “Replace, Reduce and Refine” is an important principle in research, and is incorporated in international guidelines and legislation regarding the use of animals for scientific purposes [99]. While the 3Rs were associated with animal welfare science [100], they are now increasingly implemented within mainstream scientific practice. They are now accepted as a solid ethical framework for reducing animal use and suffering. Additionally, they address public concerns on issues relating to animal research [100].

The 3Rs principle suggests that animal models that have the lowest capacity to experience pain, suffering, distress, or lasting harm be used in experiments. Thus, invertebrates (except cephalopods) are expected to replace vertebrates, whenever that is a feasible option. This is clearly based on the opinion that invertebrates have a significantly reduced, or no, ability to experience pain. The 2005 request that decapods should receive protection was challenged by UK Bioscience on the basis that sufficient, reliable evidence of pain and sentience was not available at that time [101]. Decapods are thus currently ethically preferred compared to existing protected vertebrate models, because that reduces the number of protected animals in research. However, if decapods experience pain and are sentient, a concern arises about their lack of protection and the incentive to use them in experiments [102].

Despite the exclusion of decapods from the European Directive on the protection of animals used for scientific purposes [8], there has been considerable evidence of a shift in views about sentience in this taxon. For example, the UK Government, together with several organizations, are producing a detailed animal welfare manifesto for protecting and improving crustacean welfare following Brexit [101,103,104]. The British Veterinary Association policy document on the slaughter of animals for food [105] states “*Evidence indicates that decapods (e.g., lobsters, crabs) and cephalopods (e.g., octopus, squid) are sentient, and experience pain and distress. We therefore support the principle that commercially caught decapods and cephalopods should be stunned before slaughter*”. Further, the Royal Society for the Protection from Cruelty to Animals (UK) (RSPCA) [106] states “There is currently debate about whether species like decapod crustaceans (crabs, lobsters etc.) and cephalopods (octopus, squid etc.) are sentient. The RSPCA and many others believe that there is sufficient scientific evidence to indicate that these animals should be considered as sentient, and therefore protected appropriately by legislation. This would help ensure they are no longer subjected to some of the current practices, like boiling crabs and lobster alive, that cause serious pain and distress”. This view is supported by the Humane Slaughter Association (UK), which is currently funding a research project to improve the welfare of decapods at slaughter. Concern for the welfare of invertebrates such as decapods has been expressed in recent guidelines for the ethical use of animals in field settings for research [107,108]. Further, numerous authors have supported the idea of pain in decapods, or at least the need to use the precautionary principle [102,109,110,111,112,113].

Some European countries, such as Norway, Switzerland, UK and other countries outside Europe (Australian Capital Territories, Canada, New Zealand), have already included some invertebrates in their national legislation [114] (Table 2). In Swiss legislation, decapods are protected by the secondary regulations of the Federal Food Safety and Veterinary Office through Animal Welfare Ordinance 2008, which outlines regulations for animal husbandry, transport and slaughter. Decapods are granted protection under general husbandry guidelines, and in 2018 specific protection from boiling as a means of killing was given. Decapods are included in the Norwegian Animal Welfare Act 2009 that states as follows: “*Animals have an intrinsic value which is irrespective of the usable value they may have for man. Animals shall be treated well and be protected from danger of unnecessary stress and strains*.” [115].

Despite this widespread and growing acceptance of sentience in decapods, some authors have rejected the idea of pain in crustaceans. Their objections are frequently based on confusion about the often-used phrase of data being consistent with the idea of pain [64]. This has been taken to mean that proof of pain is being claimed [116,117], despite experimental studies repeatedly stating that pain cannot be proved in any animal. Further, Diggles [118] suggests that experiments reporting findings consistent with pain are of low quality. Diggles [118] further suggests that the criteria used to judge the possibility of pain [10,102] are not suited for that purpose, and should not be used as evidence to invoke the precautionary principle. However, these critical authors have not published experimental studies that might be used as examples of high quality, and the reasons for their objections are frequently not clear. For example, they are confused about the requirement for conscious processing, first demanding that it must be demonstrated, and then a few lines later accepting that it is not possible to demonstrate consciousness [117]. They maintain that the current evidence for pain should not be used to change the way that crustaceans are used in science or fisheries/aquaculture industries [119]. These contentions have been rebutted in detail [53,119].

## 4. Conclusions and Future Directions

Given that “*all animals have an intrinsic value*”, (Recital 12, Council of European, 2010), it is necessary to improve the welfare of decapods used for experimental and other scientific purposes. They “*must be respected*”, and therefore “*treated as sentient creatures*” (Recital 12, Council of European, 2010) in the same way as cephalopods. In the light of recent evidence that decapods are probably sentient and probably have the ability to experience pain, the precautionary principle should be invoked in designing animal welfare and protection legislation, and should be applied to the entire order [102]. Birch [102] described the precautionary principle as “*Where there are threats of serious, negative animal welfare outcomes, lack of full scientific certainty as to the sentience of the animals in question shall not be used as a reason for postponing cost-effective measures to prevent those outcomes*”. Such legislative changes are needed to ensure humane treatment and experimental endpoints. Until such legislation changes are made, decapods should be humanely treated by researchers using them for scientific purposes. The welfare of decapods should be considered with respect to appropriate husbandry (such as housing, encouraging natural behavior and feeding), experimental procedures (i.e., restraint, anesthesia and euthanasia) and the skills/competence of personnel.

It is clear that we need guidelines on decapod care and maintenance, and the conduct of ethical and humane research involving these species, similar to those developed for vertebrates and cephalopods. Species-specific requirements are required on the following: (i) supply, capture and transport; (ii) environmental characteristics and the design of facilities (e.g., water quality control (pH, O_2_, CO_2_, salinity), lighting requirements (wavelengths and intensity of lighting)); (iii) housing, environmental enrichment, care and feeding; (iv) assessment of health and welfare (i.e., monitoring physical and behavioral signs); (v) approaches to severity assessment; (vi) disease (causes, prevention and treatment); and (vii) scientific procedures, general anesthesia and analgesia, methods of humane killing and confirmation of death. Further, sections covering risk assessment for operators and education and training requirements for carers, researchers and veterinarians would be helpful.

This does not mean that we should avoid using such animals, but that we should do so as humanely as possible in the interest of improving their welfare, which can also result in better quality outputs from experimental studies. An ethical attitude towards all animals in our care is not only good for them but also is good for us. We need to consider that on one side, the “*systematic inflicting of what we assume to be pain and suffering on other beings (or seeing it happen) can make people callous*”, [120] and that, on the other side, pain and stress can impact on other biological functions, leading to distress, which can compromise experimental results [84].

Therefore, the relevant researchers must know the biological features, life requirements and behavioral repertoire of decapods (used as experimental animals), and how to assess their welfare, the precise cause of stress and when to complete experiments (humane end-points). The inclusion of decapods in animal welfare legislation would have a number of practical implications for those undertaking research, including the supply of animals, transport, housing and handling, anesthesia, criteria for recognizing pain, suffering and distress, the application of humane end-points, and euthanasia. Such implications have been only marginally explored and, therefore, future research should address these issues. Their inclusion will be important not only to ensure an appropriate standard of welfare, but also to maintain public support for crustacean-based research.

In conclusion, we should be concerned about decapod welfare, and propose that there should be protection for these animals. While guidelines for ethical methods are welcome, there is a need to provide animal protection legislation for decapods, and to apply the same ethical rules to them as is currently provided for cephalopods and vertebrates.

## Figures and Tables

**Table 1 animals-11-00073-t001:** Criteria for pain perception [10,98].

Criteria	Species
(1) Nociceptors	M, B, A, F, C, D, I
(2) Pathways to central nervous system	M, B, A, F, C, D, I
(3) Central processing in brain	M, B, A, F, C, D, I
(4) Receptors for analgesic drugs	M, B, A, F, C, D
(5) Physiological responses	M, B, A, F, C, D
(6) Movement away from noxious stimuli	M, B, A, F, C, D, I
(7) Behavioral changes from norm	M, B, A, F, C, D, I
(8) Protective behavior	M, B, A, F, C, D
(9) Responses reduced by analgesic drugs	M, B, A, F, C, D, I
(10) Self-administration of analgesia	M, B, F
(11) Responses with high priority over other stimuli	M, F, C, D
(12) Pay cost to access analgesia	M, B, I
(13) Altered behavioral choices/preferences	M, B, A, F, C, D, I
(14) Relief learning	M, B, I
(15) Rubbing, limping or guarding	M, B, F, C, D
(16) Paying cost to avoid stimuli	M, B, F, D
(17) Trade-offs with other requirements	M, B, F, D

M = Mammals, B = Birds, A = Amphibians/reptiles, F = Fish, C = Cephalopods, D = Decapods, I = Insects.

**Table 2 animals-11-00073-t002:** List of legislation and codes of practice relevant to invertebrates used in the scientific research in different countries inside and outside Europe.

Countries	Legislation	Invertebrates Protected
Italy	Legislative Decree no. 26/2014	Cephalopods
Czech Republic	Act on the Protection of Animals Against Cruelty No. 246/1992	Cephalopods
Norway	Norwegian Animal Welfare Act (2010)	Squids, octopuses, decapod crustaceans and honey bees
Switzerland	Swiss Animal Welfare Act (2008)	Cephalopods and decapod crustaceans
United Kingdom	The Animal (Scientific Procedures) Act (Amendment) Order 1993 No. 2103. UK Government, London.	Cephalopods (octopuses)
Australia	Australian Code of Practice for the Care and Use of Animals in Scientific Procedures (2013)	Cephalopods (octopuses and squids)
Canada	Canadian Council on Animal Care - CCAC (1991)	Cephalopods
New Zealand	New Zealand Animal Welfare Act (1999, as at 08 September 2018)	Octopuses, squids, crabs, lobsters, or crayfishes

## Data Availability

Not applicable

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
