# Peer review of "Why Protect Decapod Crustaceans Used as Models in Biomedical Research and in Ecotoxicology? Ethical and Legislative Considerations"

_animals, 2021, doi:10.3390/ani11010073_

Round 1

Reviewer 1 Report

This paper builds a convincing argument for sentience in decapod crustaceans and the ability to experience pain and suffering. It challenges the alternative view that decapod crustaceans respond to noxious stimuli purely by nociceptive reflexes. Nociceptive reflexes are unvarying, yet the evidence reviewed shows decapods demonstrate behavioural trade-offs, protective motor reactions, long-term motivational changes and avoidance learning. Further evidence examined includes neural complexity, physiological responses and effects of analgesics and local anaesthetics. This evidence is used to recommend inclusion of these animals in European animal protection legislation. Some minor points; Simple Summary Line 14 ‘excludes decapod crustaceans (lobster, crab and crayfish)’. As written, it may appear that lobster, crab and crayfish are the only decapod crustaceans. Suggest adding ‘for example, lobster, crab and crayfish’ Abstract Line 23 ‘are not covered by current legislation in Europe’. This contradicts lines 290-299 and Table 2 which describe legislation in European countries that protects decapods. Write something more specific in line 23 like ‘are not covered by current legislation from the European Parliament’ Section 1 Line 71-72 ‘decapods were excluded from the subsequent 2010 EU Directive because of limited evidence for sentience in this group’ Decapods were excluded because the 2005 Animal Health and Animal Welfare Panel of the European Food Safety Authority cited studies such as those in crabs that associated a reduction in defensive responses with a specific analgesic effect of morphine when this may have been due to general non-responsiveness. It may be more accurate and flow into the next point better (‘Here we examine that position in the light of recent studies’) to write something like ‘decapods were excluded from the subsequent 2010 EU Directive due to refutation of the conclusiveness of studies cited by the Panel’. Furthermore, this would be more consistent with the account provided in lines 265-267 - ‘The 2005 request that decapods should receive protection was challenged by the UK Bioscience on the basis that sufficient, reliable evidence of pain and sentience was not available at that time.’ Section 2 Line 78 ‘it is can produce distress’ should be ‘it can produce distress’ Section 3 Line 291 ‘countries outside Europe (Australia Capital Territories, Canada, New Zealand)’. There is no country called Australia Capital Territories. The country is Australia. If referring to the territory of Australia which contains the national capital, it should be the Australian Capital Territory.

Author Response

Dear Reviewer,

Thank you very much for your suggestions. We revised the manuscript.

Reviewer 2 Report

Dear Authors,

I read the manuscript titled "Why protect decapod crustaceans used as models in biomedical research and in ecotoxicology? Some ethical and legislative considerations" with interest. The presented topic is of interest of the scientific community in general and there is no doubt that it has merit and can provoke further discussion about the welfare of decapods. The text is well-written, well-structured and I have just minor comments. These are easy to follow (e.g. corrections in some Latin names and in references). All my suggestions and comments are highlighted directly in the PDF copy. I recommend this paper for publication in the Animals after minor revision.

Sincerely,

Author Response

Dear Reviewer,

Thank you very much for your time and all your comments, we greatly appreciate your suggestions how to improve our paper. We revised the manuscript in relation to your suggestions and more detailed answers are given below. The changes made in the manuscript to address comments are highlighted in yellow.

Response to your comments

- Line 14: Not only decapods but all invertebrates except adult stages of cephalopods as is my knowledge. This should be mentioned/discussed.

We have discussed this in section 1, lines 42-46. The Directive 2010/63/EU excludes all live cephalopods (Art. 2, co. 3, lett. b).

- Line 34: Do not repeat the same term as included in the title. I suggest to replace this one with "Decapoda". Do not use plural in keywords.

We have modified as you have suggested.

- Line 44: Adults

Not only adult. The Directive 2010/63/EU states “3. This Directive shall apply to the following animals:(a) live non-human vertebrate animals, including:(i) independently feeding larval forms; and(ii) foetal forms of mammals as from the last third of their normal development;(b) live cephalopods”.

Line 51: I suggest to discuss behavioural patterns similar vertebrates in case of social taxa.

See: Patoka, J., Kalous, L., & Bartoš, L. (2019). Early ontogeny social deprivation modifies future agonistic behaviour in crayfish. Scientific Reports, 9: 4667. https://doi.org/10.1038/s41598-019-41333-8

We have inserted after varied behaviour the following sentence “…which may be modified by early social conditions” and the reference.

Lines 59: One of the most used model crayfish is Procambarus virginalis. I strongly suggest to add this species here.

See this paper and citations herein:

Hossain, M. S., Patoka, J., Kouba, A., & Buřič, M. (2018). Clonal crayfish as biological model: a review on marbled crayfish. Biologia, 73(9), 841-855.

We have added the species Procambarus virginalis and the reference

Line 60: You named incorrectly the species! You mentioned Procambarus virginalis! See my previous comment. Write the correct name and add suggested reference.

We have written the correct name and added reference, as you have suggested.

Line 75: Do you mean also eggs, larvae and juveniles? I assume that this approach is in comparison with adults hardly acceptable.

YES.

Line 86: Repeating "because" in the same sentence. Rewording required.

We have reworded the sentence.

Line 135: Generally yes but there is the exception of adult Birgus latro.

We have inserted the word “these” before hermit crabs.

Line 150: clarkii

We have modified it

Line: 157: I suggest to discuss here the above mentioned paper where learning based on social experience of P. clarkii is highlighted (as well as changes in behaviour when disrupting the maternal care):

Patoka, J., Kalous, L., & Bartoš, L. (2019). Early ontogeny social deprivation modifies future agonistic behaviour in crayfish. Scientific reports, 9: 4667.

The study by Patoka is interesting but not relevant to the section at line 157, which is about learning to avoid noxious stimuli.

Line 173: 54,70?

We have deled 54 number.

Line 174: Do you mean "americanum"?

Yes, americanum. We have modified.

Line 178: This is stomatopod not decapod!

We have deleted “…. mantis shrimp, Squilla mantis ….”.

Line 232: flipping?

No flicking.

Table 1: Delete the space.

We have cancelled the space.

Line 263: adult cephalopods?

It is referred to all live cephalopods.

Table 2: The same in the Czech Republic. Act on the protection of animals against cruelty, No. 246/1992 Coll.

We have added your suggestion in the table 2.

Line 349: Maybe: biological features, life requirements and...

We have added your suggestions

Line 357: Maybe also other sectors can be influenced: e.g. commercial and ornamental aquaculture...

Naturally, also other sectors could be influenced. But, in this context, we prefer not to include other sectors.

References: Check the references listed in detail. I highlighted certain errors...

We have checked the references and modified as suggested.

Yours faithfully,

Annamaria Passantino on behalf of all the authors.
